# Active head rolls enhance sonar-based auditory localization performance

**Lakshitha P. Wijesinghe**[1]*, **Melville J. Wohlgemuth**[2], **Richard H. Y. So**[3], **Jochen Triesch**[4], **Cynthia F. Moss**[5], **Bertram E. Shi**[1]

**1** Department of Electronic and Computer Engineering, Hong Kong University of Science and Technology, Hong Kong, **2** Department of Neuroscience, University of Arizona, Tucson, Arizona, United States, **3** Department of Industrial Engineering and Decision Analytics, Hong Kong University of Science and Technology, Hong Kong, **4** Frankfurt Institute for Advanced Studies, Frankfurt am Main, Germany, **5** Department of Psychological and Brain Sciences, Johns Hopkins University, Baltimore, Maryland, United States

* lpwijesinghe@ust.hk

**Data Availability Statement:** The code and data are available in https://github.com/HKUST-NISL/Echolocation-AEC.git.

**Funding:** This work was funded by the Hong Kong Research Grants Council's General Research Fund

## Abstract

Animals utilize a variety of active sensing mechanisms to perceive the world around them. Echolocating bats are an excellent model for the study of active auditory localization. The big brown bat (*Eptesicus fuscus*), for instance, employs active head roll movements during sonar prey tracking. The function of head rolls in sound source localization is not well understood. Here, we propose an echolocation model with multi-axis head rotation to investigate the effect of active head roll movements on sound localization performance. The model autonomously learns to align the bat's head direction towards the target. We show that a model with active head roll movements better localizes targets than a model without head rolls. Furthermore, we demonstrate that active head rolls also reduce the time required for localization in elevation. Finally, our model offers key insights to sound localization cues used by echolocating bats employing active head movements during echolocation.

## Author summary

Active sensing is a crucial aspect of an echolocating bat's auditory spatial perception. Head and ear movements frequently accompany their sonar call production and reception. The big brown bat waggles its head while it tracks the position of insects in darkness; however the role of these movements is not well understood. We addressed this question using a computational model that simulates active head rotations that resemble those reported in big brown bats. Our model autonomously learns to localize targets by actively rotating the head direction towards the target. We discovered that the active head waggles improve the localization accuracy, particularly in the vertical dimension.

under grant 16213617 (LPW and BES). Also the following research grants to CFM: NSF IBN-0111973 and IOS-1460149, AFOSR FA9550-14-1-039, ONR N00014-12-1-0339. MJW was supported by the Comparative and Evolutionary Biology of Hearing institutional training grant T32 DC000046 from the National Institute of Deafness and Communicative Disorders of the National Institutes of Health awarded to A.N. Popper. The funders had no role in study design, data collection and analysis, decision to publish, or preparation of the manuscript.

**Competing interests:** The authors have declared that no competing interests exist.

## Introduction

The perception-action cycle is a circular process underlying any agent's interaction with its environment. Environmental stimuli generate sensory input, which is transformed into an internal representation of the environment and forms the basis of motor action that modifies the agent's relationship to the environment, altering the sensory input, and so on. In this paper, we define perception narrowly as the process by which an organism transforms its sensory input into an internal (neural) representation. Perception is often viewed as a passive process, but the active nature of perception is becoming increasingly appreciated, primarily in the context of vision [1, 2]. Active perception refers to the process by which agents actively control their sensory apparatus to improve the quality of the perceptual representations. Common examples in vision are eye and head movements. A closely related phenomenon is active sensing, where organisms probe the environment with self-generated energy. Key examples include electrolocation by electric fish and echolocation by bats and dolphins. Both active perception and active sensing recognize that agents can control aspects of their sensory input in order to enhance or sharpen perceptual representations. Both are growing areas of research in biology and in robotics.

The goal of this paper is to unravel the contribution of active head movements to auditory localization. Here we focus on bats, nature's auditory experts, which exhibit both active sensing and active perception. Active sensing in bats can be categorized into two main strategies. First, they emit ultrasonic calls and use the reflected echoes to detect and localize objects around them. Second, they layer a wide range of behaviors on top of the call-echo behavior, such as active control of head and pinnae positions, to improve detection/localization [3, 4]. This paper considers the contribution of the echolocating bat's active head movements to auditory localization.

Different bat species utilize a variety of actions to enhance sonar localization. Bats in the *Rhinolophid* and *Hipposiderid* families move the pinnae rapidly in the anterior and posterior directions during call production and reception [5]. These movements of the pinnae create interaural intensity differences and Doppler shifts in the constant frequency (CF) signals used by these bat species, which help them localize sound sources, particularly in the vertical plane [6]. Bats that rely on frequency modulated (FM) signals, such as the big brown bat (*Eptesicus fuscus*), actively control their head and ear orientation while tracking an insect in darkness [7]. Big brown bats emit sonar sounds through the open mouth, but other species, such as nose-leafed bats, emit sounds through the nostrils. Nose-leafed bats adapt the shape of the motile leaf nose by bending and stretching actions that are temporally correlated with sonar call emissions [8]. The call-correlated movements are believed to improve echo-acoustic tracking and localization.

Motivated in part by past research findings, several sound source localization models have included active head movements. Head movements introduce changes in the acoustic cues used to localize a sound source [9]. The sensory consequences of the head movements can be used as a cue enabling an organism to learn the sound source direction. Past models have also utilized head movements to enhance sound localization performance. For instance, multi-step head control in azimuth enhances the sound source localization accuracy in reverberant environments [10]. Head movements also reduce front-back sound source confusion in challenging acoustic scenarios [11, 12]. Experiments on human sound localization have also shown that sound source localization accuracy improves as humans actively direct the head towards a target [13].

This paper advances knowledge in several ways. First, past work typically considered only one or two head rotation axes, rather than the full three dimensional rotation space considered

here. In particular, roll angles, which feature centrally here, were not included in any models we are aware of. Second, past work has relied upon predefined feature extractors, rather than learned features. Third, past work relied upon supervised learning, rather than unsupervised learning, to map auditory signals to sound source directions.

This paper presents a computational model for sonar localization in the big brown bat. Through this model, we seek to elucidate the potential contribution of head movements to sound localization accuracy. The perched big brown bat waggles its head while tracking a tethered insect [7]. Head waggles change the relative elevation between the two ears. We model this behavior as a rotation about the head roll axis, and investigate (1) the extent to which these rotations enhance the bat's localization performance, and (2) the mechanisms underpinning this enhancement.

In our model, the head orientation is controlled around multiple axes through yaw, pitch and roll rotations. The active efficient coding (AEC) framework is used to concurrently learn an auditory representation and a head control policy to localize the sound source as the bat behaves in the environment. The AEC framework extends the classic efficient coding hypothesis [14, 15] to active perception. The AEC framework has been previously applied to applications in vision, such as learning of eye vergence control [16] and smooth pursuit control [17]. Here, we demonstrate the first successful application of AEC to auditory localization.

Our model is developmental. At the start the bat behaves randomly, but over time, through exposure to received echoes, it learns a neural representation that efficiently encodes the auditory signal it hears. Concurrently, it also learns how to map the neural representation to actions that control head yaw and pitch to direct its head towards the target through reinforcement learning. Both the neural representation and the behavior are learned without external supervision or knowledge of the ground truth target directions.

When we introduce random head rotations around the gaze direction, our model exhibits improved target localization performance. Improvement increases with the magnitude of the rotations, because larger rotations enable the bat to sample the incoming echoes from a greater diversity of acoustic viewpoints, much in the same way that saccadic eye movements change an observer's optic viewpoints of the environment. Thus, our model has important implications not only for unravelling sound localization processes in bats and other mammals, including humans, but also in building a better understanding of shared principles governing the development of sensorimotor control in general.

The remainder of the article is organized as follows. In the Materials and methods section, we show the statistics of bat's head roll movement quantitatively. Then, we propose a system architecture to autonomously learn auditory localization using multi axis-head rotation. The Results section illustrates the performance gain on localization with the active head rolls. Finally, we discuss our work relative to past research on bat echolocation models, along with behavioural and neural evidence reported in the bat echolocation literature.

## Materials and methods

### Ethics statement

All procedures employed were approved by the University of Maryland Institutional Animal Care and Use Committee, protocol number R-13-04, and the Johns Hopkins University Institutional Animal Care and Use Committee, protocol number BA 14A111.

### Modelling 3D head pose during echolocation

To compute the distribution of 3D head poses during sonar localization, we analyzed the head poses recorded from the big brown bat (*Eptesicus fuscus*) while it tracked a moving

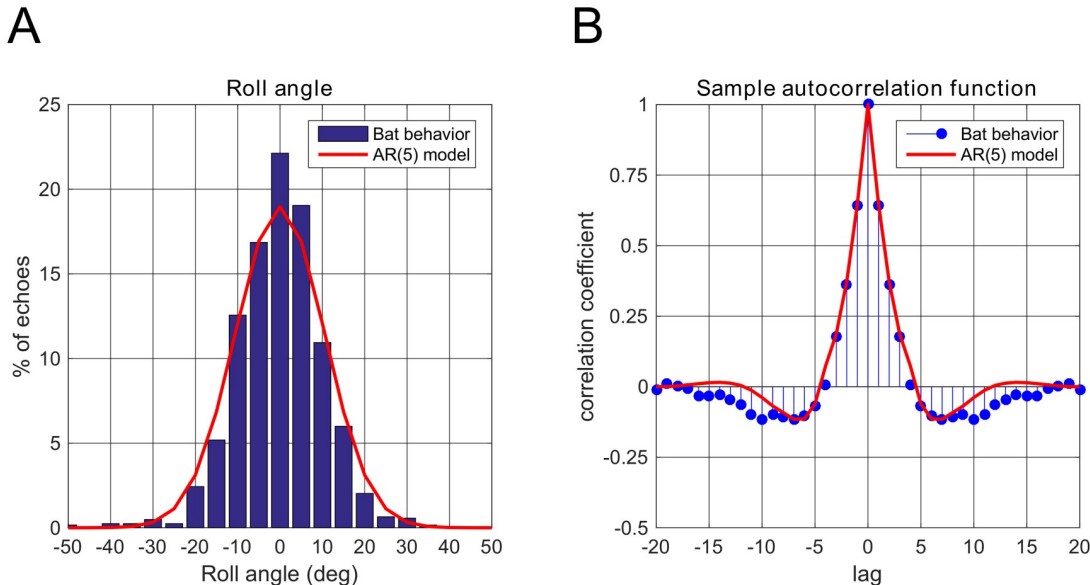

**Fig 1. Statistics of the head roll angle.** (A) Histogram of the roll angle during behavior (blue) and the marginal distribution of the fitted AR(5) model (red). (B) Sample auto-correlation function (blue) and the fitted AR(5) auto-correlation function (red).

target in the dark [7]. In this experiment, three bats were trained to rest on a perch and track a target (tethered insect) as it approached the bat, starting from a 2.5 m distance. Three high speed motion capture cameras recorded the location of markers placed on the bat's left and right pinna tips and the head. These were used to compute head position and orientation in 3D coordinates. During the experiment, the bat emitted a series of sonar calls, which were also recorded at a sample rate of 250 kHz. The return time of the echoes was computed using the velocity of sound and the known distance between the bat and the target. See also [7].

We computed the azimuth, elevation and roll angles in the Fick gimbal system from the points recorded from the bat head (see S1 Fig and S1 Text). We focused on the 3D head poses at the time of echo arrival in our analysis. To extract the head pose during localization, we selected only the data corresponding to echos received while the target was in the range 1 to 2.5 m, as this range corresponds to the range in which bats seek to localize targets. Fig 1A shows the distribution of head roll angles computed over 1234 echoes recorded for the three bats. The standard deviation of the roll angle data is ~10 deg. The sample autocorrelation function of the roll angle is shown in Fig 1B.

To emulate the real data, we sampled roll angles from an echo-to-echo autoregressive (AR) model, where each sample corresponds to a received echo. The parameters of the AR model were obtained using least squares fitting of the AR model's autocorrelation function to the sample autocorrelation function. To avoid overfitting, we chose order of the AR model to be five based on cross-validation experiments where we swept the modelling order and evaluated the fit on validation data separate from that used to estimate the AR model parameters.

## System architecture

Fig 2 illustrates the system architecture of our developmental model for echolocation. The bat actively controls its head orientation to localize the target in horizontal and elevation angles $(\alpha, \phi)$ in allocentric coordinates (see S2 Fig). The horizontal angle $\alpha$ is related to the azimuth $\theta$

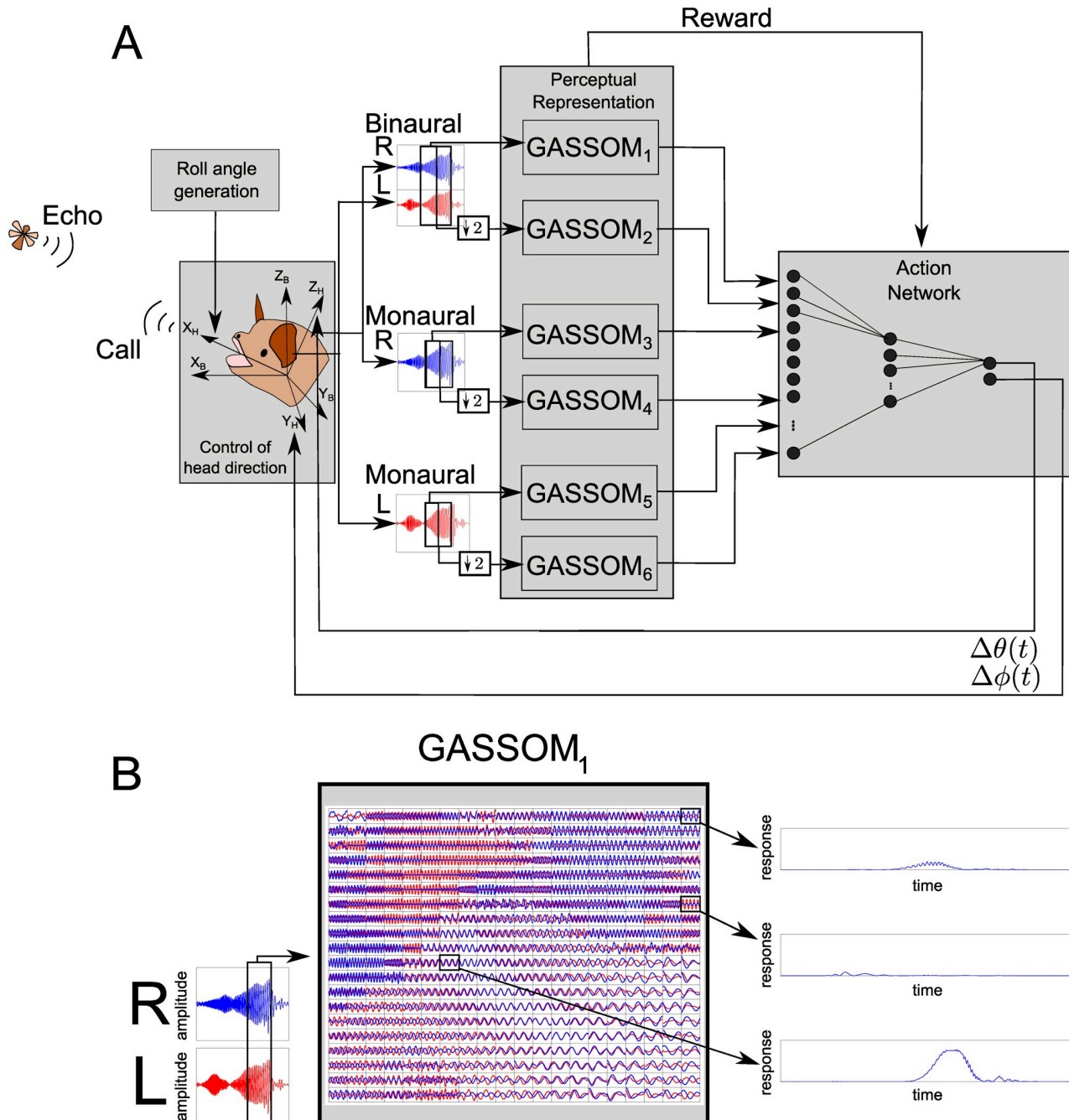

**Fig 2. System architecture.** (A) The system architecture of our echolocation model. The reflected echo is received at the left (red) and right (blue) ears. The model neural representations are obtained using six GASSOM units (GASSOM$_{1-6}$), which encode the received auditory signals at each combination of two tune scales (coarse and fine), and three aural combinations (binaural, monaural-left and monaural-right). The coarse scale windows twice as long as fine scale windows, but are downsampled by a factor of two prior to encoding. The action network maps the GASSOM responses to azimuth and elevation head control commands. An additional random head roll movement is introduced so that the bat controls the head orientation around all three axes. (B) The input and output of the fine scale binaural GASSOM. The inputs to the GASSOM are temporal snapshots from sliding windows. The GASSOM$_1$ contains a dictionary of basis vectors. The response from a few selected basis functions is shown as a function of the location of the sliding window.

and elevation $\phi$ angles according to the following equation,

$$\alpha = \sin^{-1}(\sin(\theta)\cos(\phi)). \qquad (1)$$

The bat localizes by updating the head orientation after each received echo to try to direct its head towards the target. We assume the head direction to be aligned with the roll axis (i.e. straight forward in front of the head). At the time of each echo, we also apply a random head roll movement sampled from the AR(5) process described above. To investigate the effect of introducing these roll angles, we ran simulations with roll angles with different standard deviations: $\sigma \in \{0, 10, 20, 30\}$ deg.

In our simulations, the target is assumed to be at a constant distance 1 m away from the bat head. The direction to the target is randomly chosen in angular space specified in horizontal and elevation angles ($\alpha, \phi$). We chose to use horizontal, rather than azimuth angles, because this was the space in which the Head Related Transfer Function (HRTF) data used to synthesize the received echos (see below) was sampled.

In the experiments, the target was moving towards the bat. Our model applies to the bat's behavior during localization, i.e. when the target is in the range from 1-2.5 m away. Taking the change in distance into account would not substantially change our findings, as we normalize the magnitude of the received echos across both ears simultaneously, preserving the relative spectral characteristics, but discarding overall magnitude. In the range considered, the bat emitted calls at an average pulse interval of 82.3 ms. The number of emitted calls is 16.01±6.86 across different subjects and trials. Therefore, in our simulations, we assume the target direction to be consant over 20 call-echo pairs.

One received echo and head orientation update is considered to be one iteration of our model. For each target direction, the system updates its head orientation for 20 iterations. At the beginning of training, when the system is just initialized, it behaves randomly. However, after training it hones in on the target after a few iterations. After each set of 20 iterations, a new target direction was chosen randomly.

**Auditory signal synthesis.** In our simulations, we fixed the bandwidth of the sonar call emitted by the bat. The call is a downward frequency modulated (FM) sweep from 70 to 15 kHz, which encompasses the bandwidth of sonar calls used by the big brown bat during the search, approach and interception phases of insect capture. The sampling rate of the call is 500 kHz. The duration of the call is randomly chosen from a gamma distribution $\Gamma(8.0, 0.3)$, which has mean 2.4 ms and standard deviation 0.85 ms. This was chosen to match the distribution of calls during the insect approach phase when the target is between 1.0 and 2.5 m away.

Given the call signal and the direction to the target, we synthesized received echo using measurements of the HRTFs taken from three big brown bats [18]. Measured HRTFs are available at a finite number of locations in the frontal hemisphere. At other locations, we interpolate the HRTFs. The left and right echoes are synthesized by filtering the calls by the left and right interpolated HRTFs. We assumed the target to be a point reflector and did not consider the directionality of the call emission. We also assumed that there was only one target. Although taking into account emmission directionality, greater target spatial extent and multiple targets would make the model more accurate, we elucidate in the discussion why we do not expect that it would change our main findings.

**Perceptual representation.** In the implementation of AEC used here, we model the neural representation of the binaural acoustic input using the Generative Adaptive Subspace Self-Organizing Map (GASSOM) algorithm [19]. When applied to time varying binocular visual stimuli, this algorithm learns a set of units with receptive fields and response properties similar to orientation, disparity and motion selective complex cells observed in the visual cortex. When applied to the binaural auditory stimuli here, the model learns frequency, frequency

sweep, interaural level and interaural time difference selective units. The features emerge without supervision to reflect the statistics of the incoming stimuli. Because these incoming stimuli come from different directions, the resulting representation implicitly encodes sound direction.

The left and right ear auditory signals are divided into fine and coarse (50 and 100 sample) windows with a stride of 5 samples. The coarse scale window is downsampled by a factor of 2 prior to encoding. We denote the windows by $x_{s,e,i}$, where $s$ indexes scale (c or f), $e$ indexes the ear (l or r) and $i$ is an integer index to the window. We create a binaural vector $x_{s,b,i}$ concatenating left and right ear windows,

$$x_{s,b,i} = \begin{bmatrix} x_{s,l,i} \\ x_{s,r,i} \end{bmatrix} \in \mathbb{R}^{100}. \tag{2}$$

Extracted windows are then encoded using the GASSOM algorithm. We use separate GASSOMs for each combination of $s$ and $e$.

For completeness, we describe the GASSOM algorithm briefly here. See [19] for more details.

Each GASSOM contain a number of units (400 here), each analogous to a complex cell in cortex. Mathematically, each unit corresponds to a two dimensional subspace. Intuitively, each subspace corresponds to a particular type of stimulus, e.g. sounds at a particular frequency or with a particular interaural level difference. The learning process drives the diversity of subspaces to reflect the diversity in the incoming stimuli. The output of each unit is the squared length of the project of the acoustic window onto the subspace. Units whose subspaces are closer to the acoustic vector have larger outputs. Mathematically, the basis vectors spanning the subspaces are learned so as to maximize the likelihood of the input data. Maximizing likelihood is essentially equivalent to minimizing the reconstruction error of the subspaces that encodes each window with minimum reconstruction error, subject to some temporal smoothness constraints on the encoding subspace. The reconstruction error of window ($s,e,i$) in subspace ($s,e,j$) is the squared length of the difference between the window vector and its projection onto the subspace. Mathematically, this is expressed by,

$$E_{s,e,i} = \left|\left| x_{s,e,i} - \Phi_{s,e,j}\Phi_{s,e,j}^{\mathrm{T}} x_{s,e,i} \right|\right|^2. \tag{3}$$

where $\Phi_{s,e,j}$ is a matrix whose columns contain the basis vectors spanning the subspace. Intuitively, the GASSOM performs a soft assignment of each acoustic window to one of the subspaces. Minimizing the reconstruction error brings each subspace closer to the acoustic windows assigned to it.

For each window $i$, the output of the GASSOM is a 400 dimensional vector containing the squared lengths of the projections of the input window onto the subspaces $j_i$, $\left|\left|\Phi_{s,e,j_i}^{\mathrm{T}} x_{s,e,i}\right|\right|^2$.

The final output of the perceptual representation of a pair of left/right auditory signals at iteration $t$ is a feature vector $f(t) \in \mathbb{R}^{2400}$, which is the concatenation of six 400 dimensional feature vectors obtained by pooling the GASSOM outputs over time $i$ (one for each combination of $s$ and $e$). We use average pooling to pool the responses in time. The feature vector $f(t)$ is the input to the action network, which controls the bat's head direction.

**Action network.**   The action network maps the feature vector $f(t)$ from the representation stage to motor commands $\Delta\theta(t)$ and $\Delta\phi(t)$. These motor commands update the bat's head direction in yaw and pitch. We use a reinforcement learning algorithm to learn a head control policy. In our algorithm, we consider $f(t)$ to be the state and the motor commands $\Delta\theta(t)$ and

$\Delta\phi(t)$ to be the actions. This mapping is performed by an artificial neural network with a single hidden layer comprising 500 tanh nonlinear activation units.

For the reinforcement learning algorithm, no labeled motor commands are required. Rather the natural actor-critic algorithm updates the weights of the neural network in seeking a policy that maximizes a discounted sum of instantaneous rewards, which are the negative reconstruction errors of each window averaged over all scales, ears, and window indices.

$$\mathrm{r}(t) = -\frac{1}{6\mathrm{N}}\sum_s\sum_e\sum_{i=1}^{\mathrm{N}}\mathrm{E}_{s,e,i}(t) \ . \tag{4}$$

We use the natural actor-critic reinforcement learning algorithm [20]. The critic network is a single layer neural network that maps state $\mathrm{f}(t)$ to a single linear output. During training, the motor commands are sampled from a Gaussian distribution with mean equal to the output of the network.

$$\begin{bmatrix} \Delta\theta(t) \\ \Delta\phi(t) \end{bmatrix} \sim \mathrm{N}(\pi(\mathrm{f}(t)), \sigma_e^2\mathrm{I}), \tag{5}$$

where $\pi(\mathrm{f}(t))$ is mapping performed by the neural network. The parameter $\sigma_e = 10$ deg controls the exploration by the reinforcement learning algorithm. This provides the stochasticity needed by the reinforcement learning algorithm to explore the policy space. However, for a well learned policy, exploration will degrade performance. Therefore, during testing, we use a greedy policy $\sigma_e = 0$ that fully exploits the learned knowledge. Thus, our testing results can be considered an upper bound on the achievable performance by the learned head control policy.

In addition to the azimuth and elevation updates computed by the action network, we also include random head roll movements as described below.

**Control of head direction.**   We indicate the body and head-centered coordinates by B and H respectively. The directional vectors $\mathrm{h} \in \mathbb{R}^3$ and $\mathrm{s} \in \mathbb{R}^3$ indicate the head direction and sound source direction, respectively. At each iteration $t$, a call/echo pair generates action commands $\Delta\theta(t),\Delta\phi(t)$. Given the head movement commands, the net rotational matrix at time $t$-1 is updated according to:

$$\bar{\mathrm{R}}_\mathrm{B}^\mathrm{H}(t) = \mathrm{R}_\mathrm{Y}(\Delta\phi(t))\mathrm{R}_\mathrm{Z}(\Delta\theta(t))\mathrm{R}_\mathrm{B}^\mathrm{H}(t-1) \ . \tag{6}$$

The head direction at time $t = 0$ is assumed to be in the $\mathrm{h}^\mathrm{B}(0) = \begin{bmatrix} 1 \\ 0 \\ 0 \end{bmatrix}$ direction. The head direction at $t > 0$ is given by:

$$\bar{\mathrm{h}}^\mathrm{B}(t) = (\bar{\mathrm{R}}_\mathrm{B}^\mathrm{H}(t))^{-1}\mathrm{h}^\mathrm{B}(0). \tag{7}$$

The corresponding head direction angles (azimuth and elevation) in body-centered coordinates are $\theta_\mathrm{h}(t),\phi_\mathrm{h}(t)$.

We compute a mean torsion angle according to Listing's law [21] corresponding to the direction $\theta_\mathrm{h}(t),\phi_\mathrm{h}(t)$ given by:

$$\gamma_{\mathrm{list}}(t) = \frac{\theta_\mathrm{h}(t) \times \phi_\mathrm{h}(t)}{2} \ . \tag{8}$$

The mean torsion angle is perturbed by the head roll $\gamma_{\text{roll}}(t)$, which is sampled from the AR(5) model

$$\gamma_{\text{h}}(t) = \gamma_{\text{list}}(t) + \gamma_{\text{roll}}(t). \tag{9}$$

There are two reasons for using the Listing's law. First, the 3D head orientation for a given head direction in azimuth and elevation may vary depending upon the actions taken to reach the head direction. To eliminate this dependence, Listing's law defines a unique roll angle for each direction. The random head rolls are added to the Listing's head roll angle to simulate active head rolls. Second, the roll angle defined by the Listing's law optimizes motor efficiency. Although Listing's law is commonly used to model eye movements, there is evidence that Listing's law also applies to gaze-directing head movements [22].

The net rotation matrix according to the Fick gimbal system is:

$$R_{\text{H}}^{\text{B}}(t) = R_{\text{Z}}(\theta_{\text{h}}(t))R_{\text{Y}}(\phi_{\text{h}}(t))R_{\text{X}}(\gamma_{\text{h}}(t)) \; . \tag{10}$$

We use this to update the sound source direction in head-centered coordinates:

$$s^{\text{H}}(t) = R_{\text{B}}^{\text{H}}(t)s^{\text{B}}. \tag{11}$$

Note that $s^{\text{B}}$ is changed every 20 iterations.

## Results

### Bat's behavior

We simulated our model at four different standard deviations of the head roll $\sigma \in \{0,10,20,30\}$ deg, for each of the three subjects for which HRTF data was available, and over multiple trials for each combination of head roll/subject. Statistical results reported here are averaged across all subjects and all trials.

**Generated head trajectories.** To give a more concrete illustration of the behavior generated by our model, Fig 3 shows example head trajectories after learning as the bat localizes a stationary target for each head roll standard deviation $\sigma$. We varied the initial target direction in horizontal angle and elevation by 20 deg spacing over the range for which HRTF data is available. The sonar call duration is varied from 1 to 5 ms. Due to head movements, the target direction moves from its initial location in the periphery and moves towards the center as the bat controls its head to localize the target. The sonar call duration is fixed at 2 ms for each testing trial shown in Fig 3. The trajectories corresponding to different target directions converge towards a point attractor in horizontal-elevation space. The point attractor moves closer to the (0,0) direction with increasing head roll standard deviation. The target located at (0,0) direction in head-centered coordinates indicates that the bat head is directly oriented towards the target. Hence, the bat's behavior changes with the head roll standard deviation. Larger head rolls result in improved localization performance.

**Effects on localization time.** In this section, we quantify the changes in the time required for the bat in our model to home in on the target for different roll standard deviations.

We obtained a measure of the time required by fitting the following function to the trajectories for targets initially located at top, bottom, left and right extremes of Fig 3.

$$s(t) = Ae^{-t/\tau} + B. \tag{12}$$

The time constant $\tau$ provides a measure of the time to control the head direction toward the target (see S3 Fig).

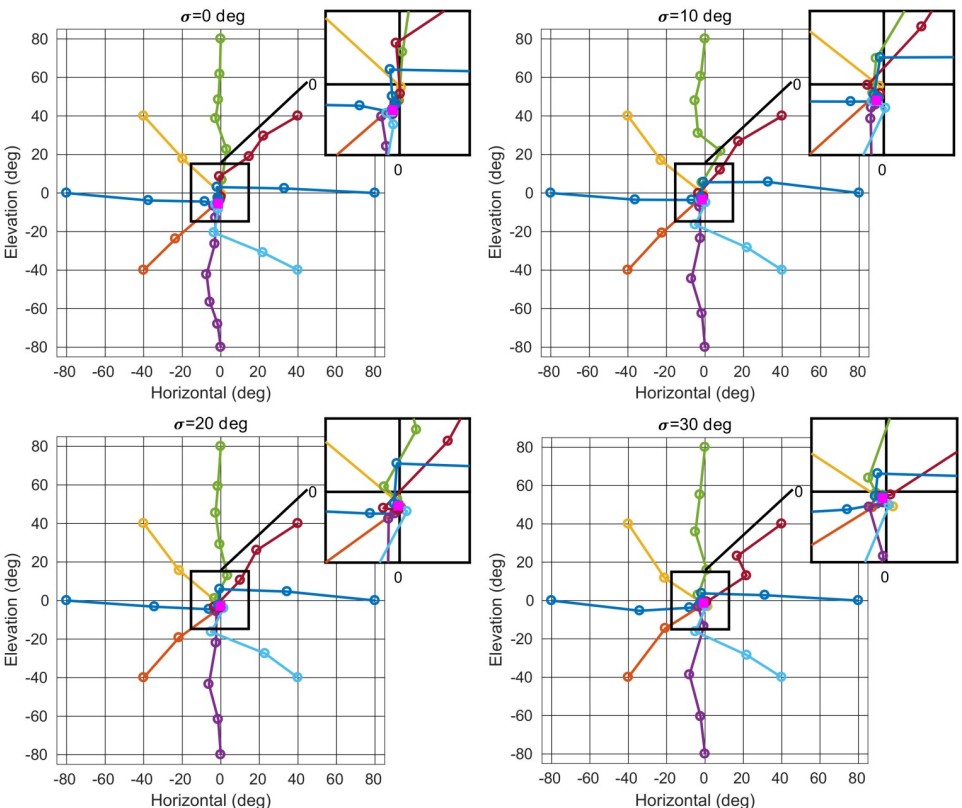

**Fig 3. Example trajectories.** Each figure shows examples of target direction in head-centered coordinates as the head direction is controlled by our model. Different colored trajectories illustrate eight initial target directions. Perfect localization corresponds to bringing the target direction to (0,0). Each circle marker indicates the target direction at iteration $t \in \{1,...,20\}$. The target direction converges to a point attractor (solid square). The inset shows a magnified view of the center region. The point attractor moves closer to (0,0) direction as the standard deviation of head roll $\sigma$ increases.

Fig 4A shows the time constant $\tau$ in azimuth and elevation directions. Each time constant is normalized by the reference time constant at $\sigma = 0$. The time constant in the azimuth direction is invariant across head roll angle standard deviation. However, in the elevation direction, the time constant decreases significantly with $\sigma$ as determined by the one-way ANOVA ($F(3,2156) = 95.05$, $p<0.001$). These results indicate that the head rolls improve the head movement speed significantly in the elevation direction, but do not degrade azimuthal control performance. The time constant is the amount of time it takes for the error to reduce by 63%. We do not measure the time required to reach a fixed accuracy, because this depends upon the initial error, whereas the time constant is independent of the initial error, as long as the controller is operating in its linear range. Nonetheless, lower time constant do correspond to lower times required to reach a behaviorally relevant accuracy given the same initial starting error.

**Effect on steady-state error.** To quantify the localization accuracy, we averaged the steady state error across the generated trajectories. The steady state error is the discrepancy between the target direction and the stabilized head direction. We compute this error by comparing the target direction in head-centered coordinates with the (0,0) azimuth and elevation direction.

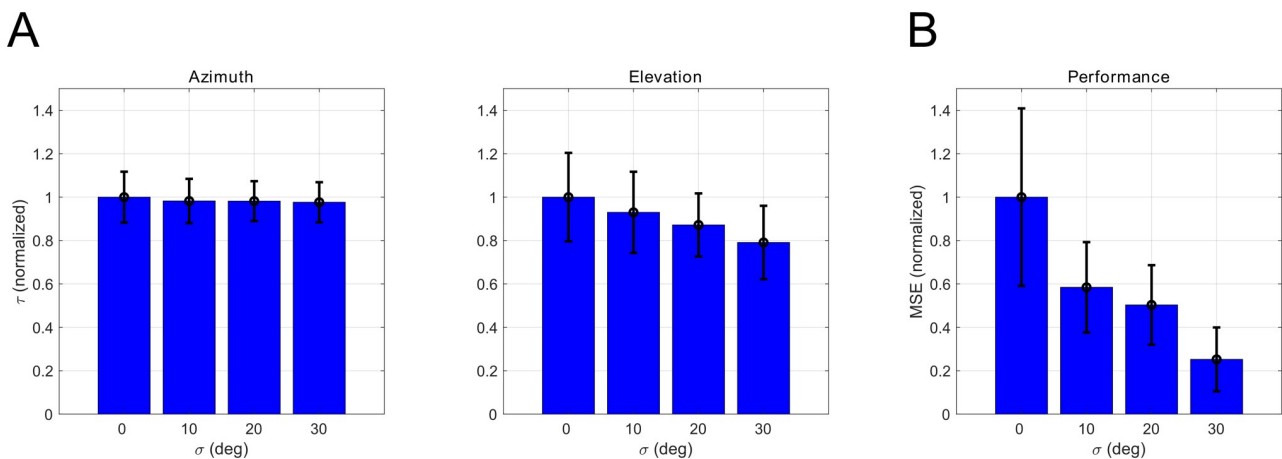

**Fig 4. Summary statistics of generated trajectories.** (A) The normalized time constant of the trajectories in azimuth, elevation direction. (B) The normalized mean squared error of the steady state head direction.

For all the test trials, the head direction stabilizes within few ($<$20) iterations. We evaluate the Mean Squared Error (MSE) at the final iteration T = 20,

$$\text{MSE} = \frac{1}{N} \sum_{i=1}^{N} (\theta_i^2(\text{T}) + \phi_i^2(\text{T})). \tag{13}$$

Here, $N = N_c \times N_{init}$ is the total number of testing trials, $N_c$ is the number of calls used for testing and $N_{init}$ is the number of target directions used for testing. Fig 4B shows the MSE at each head roll standard deviation. Each MSE value is normalized by the reference MSE at $\sigma = 0$. The results show that the MSE decreases significantly with head roll standard deviation as determined by the one-way ANOVA (F(3,32) = 13.04, p$<$0.001).

## Learned perceptual representation

In our model, the perceptual representation is described by the learned dictionary of basis vectors (see S4 Fig) spanning the subspaces used to encode the auditory stimuli. The basis vectors correspond to the auditory receptive fields that are observed in the mammalian auditory pathway. It is also important to note that they are learned during the behavior.

The basis vectors of the model can be categorized as either binaural or monaural. A few examples of the binaural and monaural basis vectors are shown in Fig 5A. The binaural basis vectors show interaural properties encoding cues commonly thought to be exploited in echolocation, i.e. Interaural Level Difference (ILD) and Interaural Time Difference (ITD).

We quantified the properties of the basis vectors by least squares fitting a linear FM sweep to the left and right components of the basis vectors (see S5 Fig and S1 Text). The amplitude difference between the two components of the binaural vector is defined as the Interaural Level Difference (ILD). The Interaural Time Difference (ITD) between the two components is computed using the phase difference between the two components.

Fig 5B shows the distribution of the ILD and ITD properties. The majority of binaural basis vectors are tuned to zero ILD and ITD values. This is to be expected, as once the localization behavior is learned, the target is usually nearly directly in front of the bat. This bias in the statistics of the auditory data shape the ITD, ILD of the binaural bases so that they are closer to zero. We compared the ILD distribution of the binaural basis to the ILD distribution of cells in the inferior colliculus of the bats. The KL divergences between the distributions are 0.1727

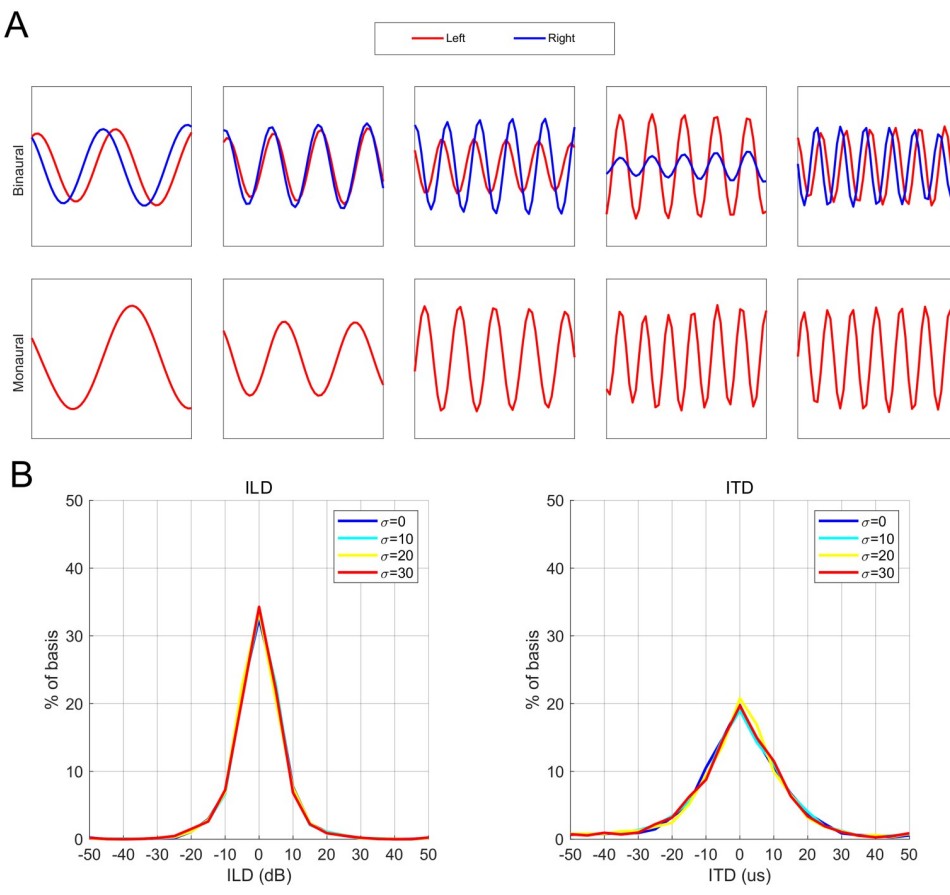

**Fig 5. Basis vectors in the perceptual representation.** (A) Binaural and monaural basis vectors. The binaural basis vectors comprise left and right (red and blue) components. The monaural basis vectors corresponding to the left (red) monaural GASSOM are shown here. (B) The distribution of interaural level, time differences of the binaural basis vectors. The mean percentage of the basis vectors are shown.

[23] and 0.2622 [24]. As a reference, we computed the KL divergence (0.782) between our distribution and a uniform ILD distribution. As shown by the KL divergence, our distribution is closer to a distribution seen in cells of bat inferior colliculus.

The ILD and ITD property distributions are similar for different head roll angle standard deviations. To quantify the similarity between the distributions, we compute the pairwise Bhattacharyya coefficient. We found all pairwise coefficients to be larger than 0.97. In addition, both the binaural and monaural basis vectors exhibit a wide range of frequency tunings, spanning the bandwidth of the call (see S6 Fig).

## Comparison with biological sound localization mechanisms

In this section, we compare the responses of the basis functions to the neural responses in mammalian auditory pathway and show that the two are similar. Typically in mammals, neurons in the Lateral Superior Olive (LSO), Medial Superior Olive (MSO) and the Inferior Colliculus (IC) show selectivity to interaural differences [25]. Specifically, in bats, neurons in the auditory pathway respond to interaural differences.

To study the interaural characteristics of the binaural basis vectors, we present the system with sinusoidal left and right auditory stimuli. First, the Best Frequency (BF) is determined for

all the binaural basis vectors. The frequency of the sinusoid is varied from 10 to 70 kHz in 1 kHz steps. The amplitude of the sinusoid is set to 1, which yields 0 ILD. The phase difference between the sinusoids is set to 0, which yields 0 ITD. The frequency with the maximum peak response is considered as the BF. Second, the sinusoidal frequency is set to BF and the ITD is set to 0. Then the ILD is varied from -40 to 40 dB in 1 dB steps. Third, for each binaural basis vector, the frequency is set to BF and the ILD is set to 0. ITD is varied from -80 to 80 us in 1 us intervals. The response in each case is normalized to have a maximum of 1. The pure tone responses are shown in S7 Fig.

The majority of neurons in the bat Inferior Colliculus exhibit monotonically changing responses with Interaural Level Differences [23, 24]. In our representation, we found a majority of 66.9 ± 3.22% of basis show similar monotonically changing responses. In addition, we found peaked responses as shown in S7 Fig. In the bat Inferior Colliculus, these type of responses are limited [23, 24].

In the mammalian auditory pathway, cyclic responses are often observed for stimuli with interaural time differences [25]. We presented pure tone responses with different phase differences which corresponds to -80 to 80 us. We found that 80.86±4.59% responses are cyclic responses. Several examples of such responses are shown in S7 Fig.

### Shaping of the reconstruction error

To understand the mechanisms underlying the improvement in the head control policy, we analyze the average reconstruction error surface at each head roll standard deviation. The learned head control policy maximizes the long-run discounted reward [20]. The reward is the negative value of the average reconstruction error as given by Eq 4. Hence, the spatial distribution of the reconstruction error is the key to understand the reason for the observed behavioral differences.

To illustrate the reconstruction error variation, we compute the reconstruction error for echoes arriving from different target directions. We vary the target direction from -80 to 80 deg in steps of 2 deg in both azimuth and elevation. The reconstruction error for a given direction is averaged across multiple sonar call durations (30 calls). Fig 6A shows the reconstruction error corresponding to -20, 0, and 20 deg head roll angles. Head rolls rotate the reconstruction error surface around the (0,0) direction.

Fig 6B illustrates the reconstruction error surface resulting from the introduction of head rolls for four roll standard deviations. To obtain this surface, the reconstruction error for a given direction is averaged across 30 call durations and 100 head roll angles. Fig 6B shows that the error surface varies with the roll standard deviation. As the roll standard deviation increases, the location of minimum in the error surface shifts more and more towards (0,0). We also observe a steeper increase in reconstruction error as the elevation moves away from zero. Since the reinforcement learning seeks a policy that minimizes reconstruction error, this shift in the minimum towards (0,0) makes the learned localization policy more accurate, especially in the elevation.

### Comparison with a system incorporating a cochlear model

In this section, we compare the performance of our system, where the GASSOM is applied directly to windows of the acoustic waveform with a system where the GASSOM is applied to the output of a cochlear model. We use the cochlear representation proposed in [26, 27]. We briefly describe the implementation of the model in this section. In the implementation, we use 81 band pass filters from 20 kHz to 100 kHz to filter the raw auditory signal. The filtered output is then sent through a half wave rectifier and a low pass filter with a cutoff frequency of

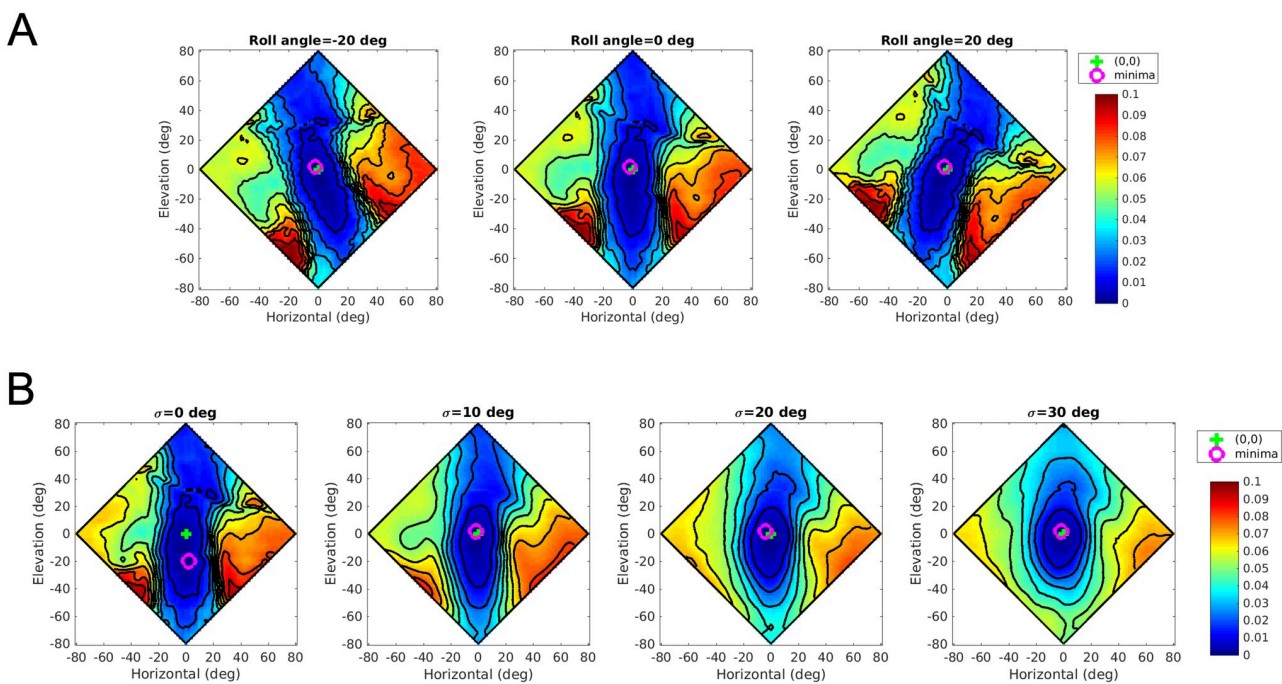

**Fig 6. The reconstruction error surfaces.** The reconstruction error as a function of horizontal and elevation coordinates. The symbols "+" and "o" indicate the (0,0) coordinates and the minima of the error surface respectively. (A) The reconstruction error for the GASSOM ($\sigma$ = 10) at the head roll angles of -20, 0, and 20 deg. (B) The reconstruction error surface at different head roll standard deviations.

1 kHz. The output of the low pass filter is the cochlear representation that is also the input the GASSOM in place of the raw auditory signal used before.

The cochlear representation contains spectrotemporal information extracted from the auditory data. We extract time overlapping spectrotemporal windows from this representation. The time duration of the window and the stride is set to 100 and 10 us respectively. The window duration and the stride are similar to the raw signal based model. Each window is then downsampled by a factor of 10 to generate a 810 dimensional feature vector.

In system with the cochlear model, we used three GASSOMs for binaural, left and right monaural data. The responses from the GASSSOMs are pooled over time using average pooling to map them to the head control commands. Both the representation and the action network are learned to jointly minimize the reconstruction error in encoding.

S8 Fig shows the comparison of localization performance for systems with and without the cochlear model. Our main finding, the improvement of steady state error with active head rolls, is found in both cases. To quantify the improvement, we compare the MSE at pairs of head roll standard deviations (i.e. $\sigma$ = 0 and $\sigma$ = 30). The Cohen's d effect size is smaller for the system with the cochlear model than for the system without (0.8363 < 1.7197). The system with the cochlear model did not exhibit faster localization in the elevation direction, as we observed for the syste without the cochlear model (Cohen's d effect size 0.0365 < 0.6839). We attributed the differences to the relatively impoverished representation provided by the output of the cochlear model. The GASSOM basis vectors for the system with the coclear model only respond to ILD, not ITD.

## Discussion

In this work, we propose an echolocation model that enables an agent (the bat) to learn to guide its head towards a target. Our model is based on the Active Efficient Coding (AEC)

framework, an extension of classic efficient coding ideas [14, 15] to active perception. Previous AEC models have been restricted to the visual domain, where they have explained the self-calibration of active binocular vision, active motion vision, the control of accommodation and torsional eye movements and combinations thereof [16, 17, 28, 29]. Moreover, AEC models have been developed to learn eye hand coordination tasks such as autonomous tracking of a robot arm [30, 31]. Here, we have demonstrated the first successful application of AEC to auditory localization. The success of AEC models in both the visual and auditory modalities highlights AEC's potential as a general theoretical framework to explain the autonomous self-calibration of active perception systems.

Previous work has hypothesized that head movements contribute to auditory localization. Behavioral experiments have revealed that perched bats use head waggles when localizing an approaching target [7]. Also, in obstacle avoidance, the CF bats increase their head movements when the pinnae are immobilized [6]. Head tilting in the lateral direction has also been proposed as a mechanism to generate elevation dependent cues [32]. Our modeling effort supports this hypothesis. We demonstrate that rotating the head around multiple axes improves the model's localization accuracy. More specifically, conventional head control in azimuth and elevation directions, combined with active head rolls, enhances sound source localization performance. To our knowledge, this is the first work to demonstrate that active head rolls improve the localization performance of a sound localization model. The bat head pose measurements also indicate that the bat changes the head pose using azimuth, elevation, and roll angles during sonar target tracking.

Our model does not require the explicit knowledge of the target azimuth, elevation direction for learning. Previous sonar-based echolocation models often use supervised learning techniques to learn the model parameters, where the azimuth and elevation angles of the target are assumed to be known to the learning agent [26, 27]. However, this knowledge is unavailable in the natural environments in which bats develop. Rather, young bats learn echolocation autonomously through sensory-motor feedback, i.e. observing sensorimotor contingencies arising during the perception action cycle. Hence, our model is a more plausible model for development than supervised learning based models.

The active head rolls enhance the localization accuracy of the learned policy. As shown in Fig 3, introducing head rolls shifts the point attractor of the learned policy towards the (0,0) direction. Bats are auditory experts and require high localization accuracy for survival. For instance, they lock their sonar beam axis to a selected target with a 3 deg standard deviation in the azimuth direction [33]. They also develop sound source localization skills autonomously starting from a very early age. For example, infant bats emit poorly coordinated sonar calls within few days after birth for echolocation purposes [34]. Our model explains how developing bats can self-calibrate the localization process.

Active head rolls rotate the axis determining the relative vertical offset between the two ears. This results in alternate acoustic viewpoints of the same environment. In our representation, the majority of binaural basis vectors are tuned to zero interaural level difference and zero interaural time difference. Hence, given this population of basis vectors, the echoes arriving from 0 horizontal angle are easier to reconstruct. From Fig 6A, for a fixed elevation, the error has a minimum near 0 horizontal angle. The active head rolls rotate the error function around the origin. As a result, the axis giving rise to interaural level and time differences rotates.

Visualizations of the reconstruction error surface as a function of sound source direction provide insight into the reasons for the observed gains. Fig 6B shows that the minima shift towards the (0,0) direction as the head roll standard deviation increases. When the standard deviation increases, the bat observes targets from a larger diversity of acoustic viewpoints.

When integrated over time via reinforcement learning, this diversity of acoustic views reshapes the error surface so that the direction with the minimum error is in the (0,0) direction. We emphasize that the averaged reconstruction error surface is not computed explicitly by the agent. Rather viewpoint diversity influences the shape of the value function used by the reinforcement learning agent in learning a policy that minimizes the discounted sum of future rewards. It is also important to note that there is a complex interplay between the behavior and the perceptual representation. The reconstruction error surface is changed not only due to the diversity of acoustic viewpoints, but also due to the changes in the perceptual representation (i.e. changes in the GASSOM basis functions) driven by behaviorally induced changes in the input statistics.

There are interesting parallels between the head control behavior studied here and saccadic eye movements. Saccadic eye movements change the viewpoint of the environment rapidly. Often driven by attention, they align the high-resolution foveal region of the eye towards different areas of interest, enabling an agent to gather a more complete picture about the environment [35, 36]. In our model, the bat localizes a sonar target by orienting the head towards the target. The roll head movements change the acoustic viewpoint and provide richer cues for target localization. This suggest that despite their differences, vision and audition may share similar active sensorimotor control principles.

Although we do not examine this effect here, we note that active movements may also perturb sensory representations. For example, eye movements can introduce blur. Movements of the pinna during echo reception can modify the spectral content of the received echos. An interesting follow-up of this work would be to examine whether the AEC framework could be extended to the control of such movements. Such an extension would likely require the incorporation of a predictive module, e.g. as we have discussed in [31].

The standard deviation of the head roll angles measured from the data we have collected from bats is 10 deg. However, our experiments show that further performance gains accrue at larger standard deviations. This raises the question of why observed head movements in bats is not larger. We suggest several reasons. First, the range of head movements may be limited naturally by musculoskeletal constraints. Second, the bat's movement may have been restricted due to our experimental setup, where the bat is tracking a target from a perch. Third, the bat's sonar localization behavior may have been altered by the experimental setup in some way unknown to the authors. Finally, we note that in our model, active head rolls are most critical during learning, and the bat subjects in our empirical studies were adults. Thus, one experimental prediction resulting from our model is that young bats in which echolocation is not fully developed will exhibit greater head roll standard deviation if not limited by other factors.

There are several simplifications in our model made for the sake of tractability, which, if modified, would make the model more realistic, but would not in our opinion change the primary findings. First, we did not consider the directionality of the call emission. This would change the statistics of the received signals, and thus the precise shape of the basis functions. If the call emission direction is held constant in the forward direction, then taking the spatio-spectral properties of emission into account would result in a systematic change in the received acoustic signals that could be modelled by systematic changes in the HRTF. We expect that changes in the basis would be small, since once the localization behavior is learned, the target location will be biased to lie mostly directly in front of the bat, which reduces the variability of the received signals. If the call emission direction varies, then this would introduce some variability in signals received from a given target location in head centered coordinates. Even in this case, we would expect that changes in the basis function to be small, for the same reason as above. Once localization is learned, the target lies mostly in front of the bat, so we would expect

most emissions to be directed in the forward direction. The bigger effect would be on the behavior of the model, which does not take into account changes in the emission direction. If the controller had access to the emission direction (as we would expect in the real bat), it could potentially compensate for these changes. If it did not, then changes in emission direction can be considered as a noise source, which we expect would degrade the speed of localization.

Second, our simulations contain only a single target, which is modelled as a point reflector. Echoes received from objects with larger extent and at different distances can be modeled as a superposition of multiple delayed echoes. Our model covers the localization phase, where we assume the bat has selected the target and is approaching it. Thus, we expect the target distance to be decreasing so that it is generally the closest object and the emission direction is in the direction of the object which is towards the front of the bat. Here, the spatio-spectral properties mentioned above would be a benefit, as the intensity of the emission drops of with eccentricity, reducing the magnitude of echoes from other objects. In this setting, the magnitude of echoes from other objects would be reduced. Thus, we expect that the learning mechanism would still be effective. Localization accuracy would be similar, but we expect that localization speed would be degraded, as the additional objects would introduce noise, albeit with smaller magnitude than the target echo. An interesting extension of this work suggested by this discussion would be to incorporate a learned attention mechanism that could to further improve performance by focusing on aspects of the signal that are most relevant to the target, e.g. along the lines of previous work we have done in the visual domain [37].

ITD responses we report in this article show both similarities and differences with ITD responses recorded from the bat auditory pathway. In our model, the ITD responses to pure tone stimuli shown in S7 Fig are cyclic responses. The responses reaches their maxima at a biologically relevant ITD values lying in the range ±50 us. This is consistent with the Jeffress model, which proposes delay lines and coincidence detection mechanisms for sound localization [38]. The distribution of the ITD shown in the Fig 5 confirms that the ITD values are well within the biologically plausible range.

On the other hand, the majority of responses of IC neurons to high frequency phase differences have been reported to be non-responsive ITD responses [23]. Only a very few percentage of cyclic responses are reported. A recent study shows that a majority of ITD responses to high frequency onset time differences are monotonic [39]. The monotonic responses show a linear variation of the response in the biologically relevant ITD range. In addition, ITD responses show time intensity trading [39, 40].

While our major conclusions are preserved with the introduction of a cochlear model, the reduction in the advantages gained through the introduction of active head rolls with the introduction of the cochlear model may be a source of some concern. However, we note that actual amount of information that the bat can extract from the auditory signal most likely lies somewhere between the two systems we have studied here. The system without cochlear processing likely has access to more information than available to the bat, due to the contraints of neural processing, whereas the system with the cochlear model is likely to be overly simplified and impoverished. For example, our system with the cochlear model was insenstive to ITD. However, it is known that bats have the capability to distinguish microsecond range time differences through their auditory pathway [41, 42]. Thus, the two systems studied here might be conceptualized as being a "corner analysis," where a "true" model lies somewhere in between. The maintenance of of the main advantages of head movements in both cases strongly suggests that further modifications of the system here to incorporate more accurate models of auditory processing will still likely exhibit these advantages.

In conclusion, our model demonstrates that active head rolls enhance the performance of the self-calibrating model of sonar localization. Specifically, both localization accuracy and

speed improve with the introduction of head rolls. Our model also provides an explanation of the mechanism enabling the gains in performance. There are several new directions in which this research can be extended.

First, from a biological standpoint, the 3D head pose of the bat performing different tasks may provide further insight to the importance of the active head rolls in other mammals, including humans.

Second, from an engineering standpoint, our model provides insights that can guide the development of robotic applications that require autonomous and unsupervised calibration of sound source localization. In particular for social robots, having interactions with the humans is a crucial skill [43]. The human-robot interaction demands accurate localization of sound sources. For an instance, a social robot may need to direct its attention to face the user [44]. However, a robot may have physical configuration changes during its life cycle. Self-calibration of sound source localization is preferred when such robots are introduced to new environments.

Third, in our experiments, the target remains stationary during the head control behavior (for 20 call-echo pairs). In reality, the target might change the direction as well as the distance from the bat. In such a scenario, the bat tracks dynamic auditory targets solely based on auditory feedback. Furthermore, bats are capable of tracking, navigating and intercepting targets relying exclusively on auditory cues. We intend to expand our approach to demonstrate autonomous learning to track and intercept moving targets, conceptually similar to the models that have been proposed in the literature [45]. Also we intend to quantitatively compare the model's learned tracking and interception of erratically flying insects to empirical data collected from free-flying bats chasing prey on the wing.

## Supporting information

**S1 Text. Additional details for the 3D head pose estimation approach and basis vector property estimation.**
(PDF)

**S1 Fig. Estimation of the 3D head pose.** The figure shows the plane that goes through the left pinna tip, right pinna tip and the head. The vector $V_z$ denotes the unit vector through the head and the orthocenter of the triangle. In the bat experiment, the bats adjusted inter-pinna separation. They either raise the tips of the pinnae to decrease the inter-pinna separation or lower the tips of the pinnae to increase the inter-pinna separation. We define the vector $V_z$ perpendicular to the line through the pinna tips. Therefore, the pinna movements have a small effect on the estimation of head orientation. Also, they do not change the conclusion of our findings, namely that head rolls improve localization accuracy in the vertical plane. The vector $V_y$ shows the unit vector perpendicular to the plane. The unit vector $V_x$ denotes the vector perpendicular to both $V_y, V_z$.
(TIF)

**S2 Fig. Coordinate transformation.** The target is located on the frontal hemisphere and 1m away from the origin. The target direction is indicated by azimuth and elevation angles $\theta, \phi$. The horizontal angle of the same target is indicated by $\alpha$, it is defined by the intersection between the cone of confusion and the horizontal plane. The target in $(\theta, \phi)$ in azimuth, elevation space can be transformed to $(\alpha, \phi)$ in horizontal and elevation space. The axis $X_H$ indicates the forward head direction. The axis $Y_H$ indicates the ear-ear direction. The axis $Z_H$ indicates the upward direction. As the bat roll the head around the head direction indicated by $X_H$, the

horizontal plane will be rotated around $X_H$.
(TIF)

**S3 Fig. Time constant estimation.** The figure shows the target elevation in the head-centric coordinates (blue dot), with the least squares fitting (blue line) of the step response $s(t) = Ae^{-t/\tau} + B$. For the illustrated example, A = -89.90 deg, B = 0.74 deg and $\tau$ = 2.46 iteration.
(TIF)

**S4 Fig. Fine and coarse basis vectors.** (A,B) The figures show the learned dictionary of 400 fine and coarse basis vectors. The basis vectors are shown in a $20 \times 20$ grid. Each element in the grid shows the left and right (red and blue) components of the basis vector.
(TIF)

**S5 Fig. Basis vector property estimation.** The figure shows the left and right(red and blue dot) components of the basis vectors. The corresponding least squares fit of the function g($t$) to the basis vectors here shown by red and blue lines. For the given example, Interaural Level Difference (ILD) = 8.06 dB, Interaural Time Difference (ITD) = -3.40 us, center frequency($f_c$) = 64.2 kHz and sweep rate(m) = -19.94 kHz/ms.
(TIF)

**S6 Fig. Spectral properties of the call and basis vectors.** (A) Frequency spectrum of the call. The frequency range of the call is conceptually similar to that of the big brown bat. (B). The spectrogram of a 2.5 ms long call. (B). The distribution of the frequency of the binaural basis vectors. (C). The distribution of the sweep rate of the binaural basis vectors.
(TIF)

**S7 Fig. Pure tone responses of the basis vectors.** (A) Binaural basis vectors. The left and right (red and blue) components of the basis vector are shown together. (B). The frequency tuning of each binaural basis vector. The basis vectors prefer frequencies with higher responses. (C). The pure tone response to interaural intensity differences. In the figure there are peaked and monotonic response functions (D). The pure tone response to interaural time differences. The shown responses are cyclic.
(TIF)

**S8 Fig. Comparison between the performances with different (raw signal vs cochlear) inputs.** (A) Normalized mean squared error at steady state. (B). Normalized time constant in azimuth and elevation direction. The performance is compared between the two different inputs, raw auditory signals and cochlear responses (blue and red). The error bars show the standard deviation.
(TIF)

## Acknowledgments

We thank Dr. Murat Aytekin for assistance with the HRTF data.

## Author Contributions

**Conceptualization:** Lakshitha P. Wijesinghe, Melville J. Wohlgemuth, Richard H. Y. So, Jochen Triesch, Cynthia F. Moss, Bertram E. Shi.

**Data curation:** Lakshitha P. Wijesinghe, Melville J. Wohlgemuth.

**Formal analysis:** Lakshitha P. Wijesinghe.

**Funding acquisition:** Jochen Triesch, Cynthia F. Moss, Bertram E. Shi.

**Investigation:** Lakshitha P. Wijesinghe.

**Methodology:** Lakshitha P. Wijesinghe, Melville J. Wohlgemuth, Richard H. Y. So, Jochen Triesch, Cynthia F. Moss, Bertram E. Shi.

**Project administration:** Jochen Triesch, Cynthia F. Moss, Bertram E. Shi.

**Resources:** Cynthia F. Moss, Bertram E. Shi.

**Software:** Lakshitha P. Wijesinghe.

**Supervision:** Melville J. Wohlgemuth, Richard H. Y. So, Jochen Triesch, Cynthia F. Moss, Bertram E. Shi.

**Validation:** Lakshitha P. Wijesinghe, Melville J. Wohlgemuth, Richard H. Y. So, Jochen Triesch, Cynthia F. Moss, Bertram E. Shi.

**Visualization:** Lakshitha P. Wijesinghe, Melville J. Wohlgemuth, Richard H. Y. So, Jochen Triesch, Cynthia F. Moss, Bertram E. Shi.

**Writing – original draft:** Lakshitha P. Wijesinghe.

**Writing – review & editing:** Lakshitha P. Wijesinghe, Melville J. Wohlgemuth, Richard H. Y. So, Jochen Triesch, Cynthia F. Moss, Bertram E. Shi.

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
