## [Decision Letter · Decision Letter 0]

24 Jan 2021

Dear Mr. Wijesinghe,

Thank you very much for submitting your manuscript "Active head rolls enhance sonar-based auditory localization performance" for consideration at PLOS Computational Biology.

As with all papers reviewed by the journal, your manuscript was reviewed by members of the editorial board and by several independent reviewers. In light of the reviews (below this email), we would like to invite the resubmission of a significantly-revised version that takes into account the reviewers' comments.

Four expert reviewers from different fields have reviewed your work. I have also read the manuscript. All reviewers comment on the merit of your work, but there are important issues that you need to address before the work can be published.  

For example, there are technical issue that need either rectifying (e.g. reviewer 3 points out that the code as provided does not work and you need to provide/amend the HRFT component) and there are technical issues that need amending/clarifying as pointed out by reviewers 1, 2 and 3. Reviewer 2 also points out that whilst the general application of the AEC framework is highly interesting, to support your major claim that your work provides insight into bat echolocation your model should use more biologically plausible input signals and you also need to link the output of your model more closely to bat behavior ( reviewer 2 provides many detailed suggestions for how this could be done, and I found myself  also thinking how for example movement trajectories produced by the model might match movement trajectories observed in bats). Whilst I understand that in certain circumstances not everything might be possible, I encourage you to address these concerns either via model modification, data analysis or careful discussion/justification of your choice of design and/or analysis. Reviewer 1 also suggests discussion of your work with respect to previous models (and potential applications), and reviewer 2 touches on that issue as well and provides references that might be helpful to you in that context. All comments and suggestions are included in the reviewers comments appended at the bottom of this letter.

You also need to improve your reporting of statistical results (e.g. for ANOVA analyses state the details of the analysis, and report your result with F-ratios and df).

We cannot make any decision about publication until we have seen the revised manuscript and your response to the reviewers' comments. Your revised manuscript is also likely to be sent to reviewers for further evaluation.

Sincerely,

Lore Thaler

Guest Editor

PLOS Computational Biology

Wolfgang Einhäuser

Deputy Editor

PLOS Computational Biology

Reviewer's Responses to Questions

**Comments to the Authors:**

Reviewer #1: This paper presents an improved model for sound source localization in bats. The problem is well motivated, the method appears technically sound, and technical material is very clearly presented. The authors are talented writers and the prose is easy to follow.

I would encourage the authors to provide a bit more narrative structure to the paper. In the introduction, be sure to clearly articulate the goals of the paper and what the reader should expect to gain from reading it, then outline how the paper will lead them there. Within the body, explain how each component of the model fits into the whole. Because the mathematical model is quite complex with many components, the authors should create more diagrams and figures illustrating how parts of the model interact.

The discussion section seems to repeat background information more than it discusses the results of the experiments. The discussion should highlight the insights gained from the experiments. Discuss how the proposed model performs compared to prior models. The authors should also discuss the specific implications of the work for engineers designing localization systems. They allude to that as a subject for future work, but I see no reason not to address it in this paper.

Reviewer #2: I agree with the authors that the AEC framework as applied here is very interesting and shows how the mapping of acoustic cues onto task-relevant behavior could be implemented through an unsupervised learning mechanism. However, I’m not sure I concur with the author’s claim: to have gained key insights to sound localization cues used by echolocating bats employing active head movements. I believe that for this claim to be really convincing two limitations to this study should be remedied first: replace the biologically implausible input signals by more plausible ones and present evidence that the sequential localization behavior as predicted by the model is actually observed in real bats.

Regarding the input signals:

In the manuscript microphone signals are used as input to the GASSOM units, but in the real bat, such information is not available. One window-length has approx a duration of 0.1msec (fine), 0.2msec (coarse), samplerate 250kSamples/sec. However, the output of the cochlea does not vary at that high rate (LP-filter with cutoff approx 1-3kHz), so the bat’s brain could not analyse the echo signal at this temporal scale. To be biologically plausible the input should be a cochlear-like representation.

Line 154: It would be interesting to see what features these cells would learn with more biologically plausible cochlear input instead of microphone signals.

Line 277: The basis vectors are stated to correspond with auditory receptive fields in the mammalian auditory pathway. Please provide a reference for bats to support this statement and argue why, despite the implausible input signal, this would still be the case.

Regarding the sequential nature of the localization process:

Line 130: The system updates its head orientation for 20 iterations (=20 call-echo pairs). How was this value chosen? Does it have biological relevance? In reality, the target direction relative to the bat will change over calls as either the prey or the bat and usually both will move in between calls. How would that affect the proposed localization scheme?

Line 268: The MSE is evaluated after 20 iterations. It is not clear to me that this is the most sensible performance characteristic to evaluate. As the bat’s localization seems more a compromise between speed and accuracy, it seems more logical to evaluate how many vocalizations are required before the target is localized with a behaviorally relevant accuracy (3° for example). As mentioned above, the bat has to localize in a dynamic context where relative prey position is possibly changing fast (in particular at close range) so localization speed (given a certain accuracy) seems to me the foremost performance criterium. It is also not clear to me what head-roll speeds are resulting from the proposed mechanism? What measurement rate (call-echo iterations per second) is assumed?

Specific comments:

Line 28-29: Did the bat’s ears move with respect to its head in the ‘bat experiment’-measurements used here? If so, the extracted head orientation (Fig. 1) seems problematic as it would be affected by non-rigid movement of pinnae with respect to head? Indeed, from fig S1 and the explanation on line 92, it can be concluded that movement of the pinna tips will affect the estimated head orientation.

Line 60: Why use azimuth, elevation and roll angles to characterize the head orientation (re. body/world), as the literature mostly uses yaw, pitch and roll?

Line 98: The Fick gimbal system: Please define the world (=body?) coordinate system you use. Fig. S2 is somewhat ambiguous. Is the angle alpha defined by the intersection of the cone of confusion around the ear-ear axis with the horizontal plane (Eq. 1)? But if the head is rolled, what you refer to as the horizontal plane will then rotate around what you call the head direction and will no longer be the true horizontal plane, or not?

The coordinate system used most often In acoustics is: the ear-ear axis (y-axis), the up direction (z-axis) and the forward direction (x-axis), with the last two in the mid-saggital plane and defined fixed to the head. It seems this is the coordinate system the HRTF was measured in or am I mistaken?

Line 142: Please note that you left out the spatio-spectral properties of emission despite sonar being an active sensor system. Hence, as the emission system also has a frequency dependent directivity it should be combined with the HRTF to simulate biologically plausible echo signals. Fundamentally, this wouldn’t change anything but please comment on how you expect this would affect your set of basis functions?

The simulated echo signals seem to contain only a prey echo and no clutter echoes? Also, it seems that a prey echo is modeled as a point reflector or is there also a reflection filter taken into account? To judge the biological plausibility of your proposed mechanism it would be interesting to know your thoughts on how these more realistic conditions would affect your learning mechanism and how it could deal with their possibly adverse effects?

Line 184: f(t) is obtained by pooling the GASSOM outputs. Please state what kind of pooling you did: maximum pooling or something else?

Line 216: What is the reasoning behind using Listings’s law and eq. (8). It is not clear to me why the mean head roll angle should be a function of the head direction (re. body) angles?

Line 238: It is not entirely clear whether active head rolling is limited to the training phase or whether it also occurs during testing?

Line 290: Fig. 5B shows the distribution of IID and ILD tuning of binaural basis vectors. Does this correspond with what is known about the distribution of such cells in the bat brain? If (not) so, this could be indirect evidence (against) in favor of the biological plausibility of the proposed approach.

Also, one wonders to what extent the tuning of the majority of binaural basis vectors to zero IID and ILD values is due to the reinforcement learning paradigm used. As the system is trained to home in on a perceived target it will collect very few echoes with large IID and ITD values and many with small IID and ITD values (as every track takes 20 iterations, most of which will point to a direction close to the target). I would be interested in the author’s thoughts about this point?

Line 332 and 338: ‘The recent discovery that bats exhibit head waggles as they track sonar targets …’. I’m not sure the claim in line 338 is justified. At least in the context of obstacle avoidance this insight has been around for a while. See ‘Mogdans J, Ostwald J, Schnitzler H. The role of pinna movement for the localization of vertical and horizontal wire obstacles in the greater horseshoe bat, Rhinolopus ferrumequinum. The Journal of the Acoustical Society of America. 1988;84(5):1676–1679. doi:10.1121/1.397183.’ and ‘V. A. Walker, H. Peremans, and J. C. T. Hallam, One tone, two ears, three dimensions: A robotic investigation of pinnae movements used by rhinolophid and hipposiderid bats, The Journal of the Acoustical Society of America. 104 (1), July 1998: 569-579’ for a computational analysis of what cues such behavior gives rise to. In both these papers the primary focus is on pinnae movements but head waggling by bats is proposed in both as an alternative way of generating very similar cues.

Typo’s:

Line 70: also

Line 103: computer

Ref 26: something went wrong with the author list. Also, I believe a more complete and more easily available reference to the ideas discussed there would be: ‘B. Fontaine, H. Peremans, Bat echolocation processing using first-spike latency coding, Neural Networks 22 (2009) 1372–1382’

Fig. S3: in the legend: ‘Fiting’

Reviewer #3: Software:

1. I downloaded the zipped programs from https://github.com/HKUST-NISL/Echolocation-AEC, but there is a missing file “HRTF_2.mat” that prevents the program from running.

2. azimuth, elevation, and roll angles on line 60 are confusing because there are pitch, yaw, and roll in aircraft axes. Do the azimuth, elevation, and roll correspond to yaw, pitch, and roll?

3. Line 70: there are two “also”.

4. Reconstruction error is an important concept throughout the paper, but is too abstract to understand with equation (3). Can the authors provide more explanation when it first shows up in the manuscript or at least provide a citation that describes it? In the citation for GASSOM model – citation #19, there is no reconstruction error.

5. a. The information for training the action network is not enough. Line 192: The hidden layer activations are then mapped to two scalar outputs corresponding to azimuth and elevation. Are the two scalar outputs the motor command? It looks like they are in Fig. 2.

b. Line 196: During training, the motor commands are sampled from a Gaussian distribution with mean equal to the output of the network. My understanding is in training, you teach the network with input, and true output. How are the “true” values of “the two scalar outputs” determined in training?

c. Why use a different greedy policy in testing? What benefits are there?

6. Control of head direction in methods is not in Figure 2 System Architecture. Without looking at your code, is it parallel to action network? Where should it be in the architecture?

7. In figure 2, if the five empty boxes denote the other 5 GASSOM units, why not number them GASSOM 1, GASSOM 2, and so on? Or write GASSOM UNITS on the top of the first box, and write numbers in the boxes.

8. Even there are insets that zoom in at the center, it is still difficult to figure out how each trajectory converges to the center. Maybe use different color for each target?

9. What is the evidence for the conclusion on line 238 - “the active head rolls enhanced the target localization”? What is it compared to?

Reviewer #4: In this manuscript, the authors studied how active head roll movements can aid sound localization in the context of echolocation by the big brown bat. More specifically, they sought to investigate how an autonomous model can learn to align the bat’s head direction to a target of interest. This paper is very well written, addressing clearly articulated questions, with modelling well-laid out and executed, and containing rigorous analysis. The proposed model is well thought out, concordant with previous behavioral work. Overall, the study is of high quality and therefore should be published.

Some very minor comments

- Page 4: 1234? Echoes. What a nice numerical symbology coincidence

- Page 4: “assume the head assumes” reads weird

- Page 5: When first mentioned, HRTF is not defined

- According to SI style conventions, there should be a space between a numerical value and unit symbol

- Active movements may also perturb sensory representations (for vision, retinal movement blurs the sensory representation; coordinate transformations and spatial updating all require accurate sensory and predictive information). The authors may consider mentioning this briefly.

- Similarly, there may be energetic and physical constraints to head roll motion, which is not mentioned. May this be a reason why the measured

**Have all data underlying the figures and results presented in the manuscript been provided?**

Reviewer #1: Yes

Reviewer #2: Yes

Reviewer #3: **No: **The HRTF part of the model is not working

Reviewer #4: Yes

PLOS authors have the option to publish the peer review history of their article (what does this mean?). If published, this will include your full peer review and any attached files.

Reviewer #1: No

Reviewer #2: **Yes: **Herbert Peremans

Reviewer #3: No

Reviewer #4: No
---

## [Decision Letter · Decision Letter 1]

18 Apr 2021

Dear Mr. Wijesinghe,

We are pleased to inform you that your manuscript 'Active head rolls enhance sonar-based auditory localization performance' has been provisionally accepted for publication in PLOS Computational Biology.

Best regards,

Lore Thaler

Guest Editor

PLOS Computational Biology

Wolfgang Einhäuser

Deputy Editor

PLOS Computational Biology

Reviewer's Responses to Questions

**Comments to the Authors:**

Reviewer #1: The authors have addressed all my concerns with the paper. Nice work!

Reviewer #2: All my comments have been adressed.

**Have the authors made all data and (if applicable) computational code underlying the findings in their manuscript fully available?**

Reviewer #1: Yes

Reviewer #2: Yes

PLOS authors have the option to publish the peer review history of their article (what does this mean?). If published, this will include your full peer review and any attached files.

Reviewer #1: No

Reviewer #2: **Yes: **Herbert Peremans

---

## [Editor Report · Acceptance letter]

29 Apr 2021

PCOMPBIOL-D-20-02022R1 

Active head rolls enhance sonar-based auditory localization performance

Dear Dr Wijesinghe,

I am pleased to inform you that your manuscript has been formally accepted for publication in PLOS Computational Biology. Your manuscript is now with our production department and you will be notified of the publication date in due course.

With kind regards,

Andrea Szabo
